# Research Progress on the Leaf Morphology, Fruit Development and Plant Architecture of the Cucumber

**DOI:** 10.3390/plants11162128

**Published:** 2022-08-16

**Authors:** Jie Li, Jiajian Cao, Chunhua Wang, Ning Hao, Xiaolan Zhang, Mingyue Liu, Tao Wu

**Affiliations:** 1College of Horticulture, Hunan Agricultural University, Changsha 410128, China; 2Key Laboratory for Evaluation and Utilization of Gene Resources of Horticultural Crops (Vegetables, Tea, etc.), Ministry of Agriculture and Rural Affairs of China, Changsha 410128, China; 3Key Laboratory for Vegetable Biology of Hunan Province, Changsha 410128, China; 4Graduate School of Agricultural and Life Sciences, The University of Tokyo, Tokyo 113-8654, Japan; 5College of Horticulture, China Agricultural University, Beijing 100107, China

**Keywords:** cucumber, agronomic characters, phytohormone, regulation mechanism

## Abstract

Cucumber (*Cucumis sativus* L.) is an annual climbing herb that belongs to the Cucurbitaceae family and is one of the most important economic crops in the world. The breeding of cucumber varieties with excellent agronomic characteristics has gained more attention in recent years. The size and shape of the leaves or fruit and the plant architecture are important agronomic traits that influence crop management and productivity, thus determining the crop yields and consumer preferences. The growth of the plant is precisely regulated by both environmental stimuli and internal signals. Although significant progress has been made in understanding the plant morphological regulation of *Arabidopsis*, rice, and maize, our understanding of the control mechanisms of the growth and development of cucumber is still limited. This paper reviews the regulation of phytohormones in plant growth and expounds the latest progress in research regarding the genetic regulation pathways in leaf development, fruit size and shape, branching, and plant type in cucumber, so as to provide a theoretical basis for improving cucumber productivity and cultivation efficiency.

## 1. Introduction

Cucumber occupies a large cultivation area and has many economic benefits around the world, which are closely related to people’s daily lives [1,2,3]. Cucumber fruits, whether immature or mature, raw or processed, as vegetables or as sweets, are widely consumed globally. The cucumber is also a model plant for the study of the vascular system, fruit development, and plant architecture [4,5,6].

The morphology of the leaves and fruits and the plant architecture, as the key agronomic traits of cucumber, directly affect its final yield and quality [7]. In cucumber production, suitable compact plant varieties for once-over mechanical harvesting and high-density planting are favored in the adult plant stage. In fresh markets, architectural traits such as an indeterminate growth habit, no branching, powerful main stems, and moderate leaf size are desired. Proper plant architecture may boost agricultural productivity while lowering labor expenses, which is critical given the limited arable area available. The leaf morphology and plant architecture, including plant height and branches, have great influences on the fruit weight and yield. The appearance of cucumber fruit largely influences the preference of consumers, including the fruit length and diameter and the fruit shape [8]. The identification of the key genes and molecular mechanisms controlling the leaf and fruit size, as well as the plant height and branching, is critically important for the effective transformation of the plant morphology with the required structural characteristics.

In this review, we outline recent advances in scientific understanding of the genetic regulatory networks controlling the leaf morphology, fruit development, branch extension, and growth characteristics of cucumber.

## 2. Cell Proliferation and Expansion in Plant Growth and Development

Plant organs develop from the shoot apical meristem (SAM) through cell division and differentiation [9,10]. From the meristematic primordium to the ultimate, developed size, the development of lateral organs may be considered a progressive process, beginning with founder cell recruitment to the primordium, followed by two phases of growth: first, cell proliferation (cell growth associated with division), and then cell expansion (cell growth without division) [11,12]. In the early stage, cell division mainly functions to increase the number of cells, and in the later stage, cell differentiation and cell growth mainly function to increase the cell volume, thus forming the final organ size [13,14]. For example, in cucumber, a small and cordate leaf mutant exhibits alterations in both the leaf size and shape of cucumber. When compared to wild-type leaves, the mutant has fewer numbers of epidermal pavement cells [15], while a little leaf mutant, which has a small organ size, exhibited a reduction in both the cell number and cell size [16]. In the cucumber short fruit mutant, whose fruit length is significantly lower than of the WT, one can observe a larger cell area but significant fewer numbers of cells [17]. Another longer fruit mutant exhibits a larger cell area [18].

## 3. The Roles of Phytohormones in Growth and Development

The growth of plant organs is precisely regulated by both environmental stimuli and internal signals. Indeed, plant hormones play fundamental roles in organ growth [19,20]. Phytohormones are a class of small organic molecules that function as essential regulators of plant growth, and all living cells are responsive to hormones. Phytohormones have substantial physiological effects at low concentrations, influencing plant development and stress resistance, mainly including cell division, elongation, and differentiation, as well as seed germination, rooting, flowering, fruiting, sex determination, dormancy, and organ shedding [21,22,23]. A single type of phytohormone can be involved in multiple aspects of development; furthermore, multiple phytohormones can collaborate in regulating a specific developmental process. In addition to the five classical phytohormones (auxin, CK, ethylene (ETH), gibberellin (GA), and abscisic acid (ABA)), the roles of other small molecules, such as brassinosteroid (BR), jasmonic acid (JA), salicylic acid (SA), and strigolactone (SL), in plant growth and development are becoming apparent.

The leaf morphology is the end result of the activities of different phytohormones. The leaves originate from the PZ (peripheral zone) cells, which are easy to differentiate [24]. Auxin plays a key role in these processes, as only PZ cells that overlap with the periodic auxin maxima eventually develop into leaves. Leaves are the main organs involved in photosynthesis in green plants, and light regulates the leaf initiation by activating CK signaling and affecting outflow-dependent auxin gradients [25]. GAs and BRs promote leaf growth by increasing the cell proliferation and expansion. Mutants deficient in BR biosynthesis or signal transduction have a reduced leaf size [26]. ABA has a critical role in the regulation of the rate of water loss through stomatal transpiration [27]. In cucumber, a mutant that exhibits a vascular configuration and abnormal organ morphology has a high auxin accumulation in the leaf veins [28]. Meanwhile, the plants are more resistant to downy mildew and accumulate more SA. In a mango fruit mutant, the leaves have a butterfly-like shape and substantial growth defects in the mediolateral axis. This phenotype is caused by the polar transport of auxin [29].

During fruit development, plant hormones participate in cell proliferation and jointly promote fruit growth. Auxin mainly functions to promote cell elongation and volume expansion. In cucumber, the endogenous hormone contents of different inbred line cucumber fruits have been studied and it was discovered that the indole-3-acetic acid (IAA) concentration of the fruit is positively correlated with the fruit size and cell proliferation at distinct developmental stages [30]. When ectopically expressing the *CsYUC11* coding sequence in *Arabidopsis*, as the expression level of *CsYUC11* increases, the pedicel length and the IAA content increase progressively, implying that *CsYUC11* promotes pedicel elongation through auxin biosynthesis in *Arabidopsis* [31]. Enhanced expression of an auxin-synthesizing gene (*DefH9-iaaM*) in cucumber significantly enhances its parthenocarpy ability [32]. GA can promote cell expansion and participate in the regulation of the fruit shape. The expression pattern of the cucumber GA receptor gene, *CsGID1a*, has been found to be closely correlated with fruit locule formation. Silencing *CsGID1a* leads to the phenotype of fruits exhibiting abnormal carpels and locules, while the overexpression of *CsGID1a* in the *Arabidopsis* double mutant (*gid1a gid1c*) exhibits “cucumber locule-like” fruits [33]. CK plays a decisive role in the final size of the fruit after it is transported to the fruit. In cucumbers, tZR diminishes during the early stages of fruit development and is primarily engaged in cell division during the early stages of ovary development [34]. In cucumber seedlings, transient treatment with physiological concentrations of ETH stimulates cell division and alters cell division polarity [35]. ETH is involved in regulating cucumber fruit shape development. The application of exogenous 1-aminocyclopropane-l-carboxylic acid (ACC) to straight fruit promotes fruit bending in cucumber. Overexpressing *CsERF025* in cucumber leads to an increase in the bending fruit rate by enhancing the production of ETH [36]. In cucumber, the exogenous application of BR induces parthenocarpic fruit formation, by inducing cell division, whereas its inhibitor abolishes the natural parthenocarpic capacity in a parthenocarpic cucumber [37].

Stem elongation is controlled by several hormones, including GAs, BRs, auxin, and SLs. The cucumber *dwarf* mutant exhibits short internodes due to decreased endogenous GA3 levels [38]. At present, two BR-deficient dwarfing mutants (*scp-1* and *scp**-2*) have been obtained by EMS mutation in cucumber. The dwarfing phenotype of *scp-1* can be completely restored to the wild-type phenotype after exogenous BR treatment, while the dwarfing phenotype of *scp-2* can only be partially restored after exogenous BR treatment [39,40]. SLs has been proven to directly or indirectly inhibit the germination of the plant lateral buds [41,42]. However, little research has been conducted on cucumber.

In addition to their physiological functions in plant growth, some phytohormones, such as SAs, JAs, and SLs, are usually induced by biotic or abiotic stress [43,44,45]. For example, in cucumber, the content of JAs and SAs significantly increases with *T. longibrachiatum* H9 inoculation [46]. These hormone responses provide a signal pathway for environmental factors to regulate plant growth, and it can be further concluded that phytohormones widely participate in determining the growth and size of the plant organs.

## 4. Genetic and Molecular Regulation Mechanisms of Leaf Morphology

Leaves are flat, lateral appendages on plants that act as solar panels, capturing sunlight and generating glucose and oxygen. In addition, leaves can serve as interfaces for monitoring environmental cues, such as light, temperature, water, insects, and microbes [47]. Cucumber is one of the typical dicotyledonous plant species with alternate leaves. Specifically, a petiole connects a single leaf blade to the node [48]. Typically, cucumber has palmate leaves with five major veins, from the petiole at the leaf base to the leaf edge, producing lobed leaves. In recent years, several mutants with aberrant leaf morphology have been identified, and multiple genes have been mapped and thoroughly described (Figure 1).

### 4.1. Molecular Regulation Mechanisms of Cucumber Leaf Shape

(1) PID

PINOID (PID) belongs to the plant-specific AGCV III subfamily of the large AGC protein kinase family, which mediates phosphorylation/dephosphorylation by encoding serine/threonine protein kinases. The *PID* gene family regulates the phosphorylation status of the auxin efflux transporter PIN-FORMED (PIN) and participates in the regulation of *Polar Auxin Transport (PAT)* [49,50,51,52]. PIN proteins can transport auxin out of the cell in the direction of PIN [53,54], and the polarity localization of PIN is mainly in specific plasma membrane regions and is primarily controlled by PID [55,56,57].

In cucumber, *CsPID* has been identified in some leaf mutants, mainly in the *round leaf (rl)* mutant [58,59], suggesting that *CsPID* plays a major role in the leaf shape formation. *CsPID* is believed to regulate lateral organ morphogenesis by modulating the expression of the genes related to auxin transport and signaling [59]. However, the specific regulatory mechanism of *CsPID* in cucumber has not been thoroughly studied. By referring to the well-established molecular mechanism of polar auxin transport in other model crops [60], one can observe that the IAA content of the *rl* mutant is different from that of the wild type, suggesting that *CsPID* is involved in the control of polar auxin transport in cucumber. Additionally, it has been proven that the transcriptional regulation of *CsPIN1* is regulated by *DEFORMED FLORAL BUD1-PHABULOSA (**CsDFB1-CsPHB)*, thereby affecting the distribution of auxin in cucumber [61].

(2) WOX

The Wuschel-related homeobox (WOX) belongs to the homeodomain (HD) super family, which has a typical DNA-binding domain of ~60 amino acids [62,63]. *WOX* comprises a large group of transcription factors, and WOX-HD is slightly larger (~65–70 aa) because of the extension at the HD C-terminus [64]. Previous research has identified three clades of *WOX* genes based on their phylogenetic connections and conserved domains: the ancient clade, the intermediate clade, and the WUS/modern clade [62,64]. Previous research has shown that *WOX* family members are required for embryonic patterning, stem cell maintenance, and organ creation [65].

The *WOX* gene activities of cucumber are closely connected to shoot meristem maintenance and sexual differentiation, and also have a significant impact on cucumber growth, development, and production [66]. Cucumber has 11 putative *CsWOX* genes that have been identified and described, and they are also classified into three primary clades [67,68]. The expression patterns of these *CsWOX* genes reveal that a number of them are primarily expressed in distinct organs, indicating that these genes may be involved in various developmental processes [67]. In cucumber, the *mango fruit (mf)* mutant has been identified, exhibiting narrower, wrinkled, downward-cupped, and dark-green cotyledons, as well as significantly reduced blade expansion of the true leaves [69]. Map-based cloning has shown that MF encodes a *WOX1*-type transcriptional regulator *(CsWOX1),* which lacks the conserved Wuschel (WUS) box in the *mf* mutant. This study showed that *CsWOX1* may influence early reproductive organ development and modify auxin signaling in cucumber via the *SPOROCYTELESS* (*CsSPL*)-mediated pathway. In another study, it was confirmed that *CsWOX1* controls auxin transport through *CsPID1* and further plays a role in leaf vein development [29]. During later leaf morphogenesis, *CsWOX1* can upregulate the expression of *CsPID1* and participate in the establishment of a cucumber leaf vein model, depending on the auxin polar transport process [29]. *CsWOX1* can also directly activate the expression of *CsPIN2* and participate in the development of the distal region of the leaf. In addition, *CsWOX1* regulates leaf cell proliferation by forming a feedback regulation mechanism with the *CIN-TCP* transcription factor. However, *WOX1* only has an effect on the leaf width and does not affect the blade length or complexity.

(3) Other genes

*Hanaba Taranu (CsHAN)* encodes a GATA3-type transcription factor that is involved in floral organ development, SAM organization, and embryo development. In cucumber, the overexpression and RNAi of *CsHAN1* transgenic cucumber results in delayed growth after early embryogenesis and generates strongly lobed leaves, suggesting that *CsHAN1* is crucial for SAM formation [70]. *CsHAN1* has also been discovered to regulate the *WUS* and *Shoot Meristemless (STM)* pathways in cucumber and to govern leaf formation through a complex gene regulatory network. *Irregular vasculature patterning (CsIVP)* and *YABBY 5 (CsYAB5)* are two transcription factors existing in the vascular tissues, which influence the leaf shape in cucumber [28]. The leaves in *CsIVP-RNAi* plants curl downward, and the bilateral leaf edges overlap, owing to the larger major veins and increased number of secondary veins, whereas *CsYAB5-RNAi* plants have a comparable leaf phenotype. *CsIVP* can directly bind the promoter of *CsYAB5* to enhance its expression and influence the leaf shape [28]. In short, whether there are additional genes involved in the regulation of the cucumber leaf shape remains to be further investigated.

### 4.2. Molecular Regulation Mechanisms of Cucumber Leaf Size

At present, few genes regulating leaf size have been identified in cucumber. The *little leaf* (*ll*) mutant, with small leaf sizes, has been identified [16], and *CsLL* has been demonstrated to be an ortholog of *Arabidopsis Sterile Apetala (SAP)*, encoding a WD40 repeat domain-containing protein. The small organ size in *ll* can be attributed to reductions in both the cell number and cell size [16]. In addition, a *small and cordate leaf 1 (scl1)* mutant, with fewer numbers of epidermal pavement cells, has also been researched, and it has been proven that *CsSCL* leads to alterations in both the leaf size and shape of cucumber [15].

## 5. Genetic and Molecular Regulation Mechanisms of Cucumber Fruit Development

Fruits can preserve growing seeds, corresponding to the plants’ ovaries, and provide food and nutrition for people. Cucumber fruit is widely recognized for its wide variations in fruit size and shape, which are also key agronomic features affecting the crop output and external quality [71]. Four physiological stages of early development govern classic fruit morphogenesis: ovary growth, fruit set, fast cell division, and subsequent cell expansion. In particular, the fruit size and shape are two major agronomic traits influencing cucumber production and external quality, which vary widely among cucumber cultivars due to long-term selection during domestication and breeding [72]. Because of the wide variation of the fruit length, it plays a significant role in influencing fruit size and form [73]. For cucumber, appropriate fruit length is also an essential breeding aim, because it is one of the most prominent identifying factors used to define market groups in commercial production. Generally, cucumbers have simple fruit shapes (round, oblong/oval, or cylindrical). Cucumber fruit size and shape are complex traits influenced by numerous factors, including the genotype (Figure 2) and environment [5]. This study’s results indicate that the variations in cucumber fruit size and shape result from differences in the cell numbers and shape in the longitudinal and cross-sections, driven in turn by differences in the orientation, timing, and duration of cell division and expansion [71].

### 5.1. Molecular Regulation Mechanisms of Fruit Shape

A variety of QTLs regulating the fruit size (FS), shape (FSI), and fruit weight (FW) have been evaluated in cucumber, melon, and watermelon [74]. Approximately 150 consensus QTLs have been identified for these traits, and 253 homolog genes related to grain size/weight have been cloned in *Arabidopsis*, tomato, and rice through the genome-wide investigation of three cucurbit genomes [74]. The proteins encoded by these genes contain a cell number regulator (CNR), cell size regulator (CSR), Cytochrome P450 (CYP78A), SUN, OVATE, Tonneau1 recruiting motif (TRM), YABBY, and WOX.

A total of 135 QTLs affecting the fruit shape have been genetically mapped in cucumber, with some QTLs/genes already having been discovered or cloned [75]. In a QTL analysis of FS/FSI using segregating populations resulting from a hybrid between WI7238 (long fruit) and WI7239 (round fruit), the round fruit shape has been shown to be regulated by two QTLs, FS1.2 and FS2.1, which encode the tomato homologs *SUN (CsSUN25-26-27a)* and *SlTRM5 (CsTRM5)*, respectively [76,77]. *SUN* encodes an IQD family protein that contains a conserved IQD and is involved in CaM binding. The protein encoded by *SUN* is a growth regulator that results in elongated fruit and is believed to affect hormone or secondary metabolite levels [78]. The IQD is a 67-amino acid conserved region that comprises up to three IQ motifs that increase CaM binding in the presence of Ca^2+^. In tomato, after pollination for 7–10 days, SUN controls the tomato morphology in accordance with the cell division stage, and the cell elongation and division along the proximo–distal axis increase [79].

Another frequently studied gene affecting the fruit shape is the *OVATE* gene [77]. The *OVATE* gene was first identified in tomato and is assumed to be related to the fruit shape, since its mutation results in an elongated fruit phenotype, while the overexpression of *OVATE* in the pear-shaped tomato leads to round fruit [80]. The *OVATE* gene encodes a protein with a ~70-amino acid conserved C-terminal domain, which is known as the OVATE domain, and the proteins containing this domain are referred to as OVATE family proteins (OFPs) [81,82]. *OFPs* serve a variety of roles in plant growth and development by inhibiting the transcription of target genes [80,83,84]. In melon, a gene designated as *CmFSI8/CmOFP13* has been identified as controlling the fruit shape and being associated with the FSI, which is determined by two varieties: B8, with long-horn fruit, and HP22, with flat, round fruit [85]. In cucumber, 19 *CsOFPs* have been identified and distributed on seven chromosomes that can be divided into four subgroups, named OFP I to OFP IV [86]. Most of the *CsOFPs* were expressed in the reproductive organs. When *CsOFP11* was overexpressed in *Arabidopsis*, the transgenic lines showed shorter but blunt siliques, indicating that *CsOFPs* may regulate cucumber fruit development. However, there is no research on the molecular mechanism. This research may provide ideas for the regulation of *OVATE* in cucumber fruit shapes in light of the study of the gene in other plants. For example, it has been found to interact with *TRMs* and, together, they regulate the cell division patterns in tomato fruit development [84].

### 5.2. Molecular Regulation Mechanisms of Fruit Size

The fruit size is determined by the fruit length (FL) and fruit diameter (FD). At present, only a few genes controlling the cucumber fruit size have been cloned and functionally verified, namely, *Short Fruit 1* (*SF1*), *SF2*, *SF3*, and *Fruitfull1* (*CsFUL1^A^*). The *sf1*, *sf2*, and *sf3* mutants in cucumber all have short fruit phenotypes, but they encode different proteins. Among them, both *SF1* and *SF2* regulate the fruit size through epigenetic modifications. *SF1* encodes a cucurbit-specific RING-type E3 ligase that ubiquitinates and degrades both itself and *ACS2* to regulate ETH production, which has a dose-dependent influence on cell division and fruit elongation [87]. *SF2* encodes a Histone Deacetylase Complex 1 (HDC1) protein. A deficient *sf2* allele inhibits HDAC, targeting to chromatin, resulting in increased histone acetylation. *SF2* regulates the fruit cell proliferation by directing the manufacturing and metabolism of CK and the polyamines [17]. *SF3* encodes a katanin p60 subunit, which is a homolog of *KTN1* (*CsKTN1*). The shorter fruit in the mutant is caused by reduced cell numbers, as determined through histological examination. Additionally, hormone quantitation and RNA-seq analysis suggest that *CsKTN1* may regulate the fruit length by affecting the metabolic levels of auxin and GA [88]. *CsFUL1* is a MADS-box gene [18]. It directly targets the *Superman* (*SUP*), which is known to be a regulator of cell division and expansion, to repress its expression and then negatively regulate fruit elongation in cucumber [18].

## 6. Genetic and Molecular Regulation Mechanisms of Cucumber Plant Architecture

Cucurbitaceae crops progress through vegetative and reproductive development simultaneously across most of their lives. Branching, plant type, and stem growth are also important traits for cucumber breeding objectives, and several genes have been identified (Figure 3). Because the plant type has a significant influence on the crop management and productivity, it has been exposed to severe selection throughout the processes of crop domestication and development. Shoot branching is an important aspect of plant development and fitness that is connected to crop yield. Hence, it has been focused on as a selection target during domestication [89].

The *Cucumber Lateral Suppressor (CLS)* gene has been cloned, and its transcripts are found in the leaf axils, where the axillary meristem is formed [90]. The ectopic expression of *CsCLS* in the *las* mutant of *Arabidopsis* could fully complement the reduced number of axillary buds, demonstrating that *CsCLS* has a conserved function in cucumber bud initiation [90]. *CsBRC1-RNAi* (*Branched1-RNAi)* cucumber lines show increased shoot branching and decreased auxin accumulation in the lateral buds. Biochemical data show that *CsBRC1* can bind to the promoter of *PIN-Formed3 (CsPIN3)*, the auxin efflux carrier, and inhibit its expression. The high expression of *CsPIN3* can result in increased lateral branching and decreased auxin accumulation in buds [91].

Flowering plants have two forms of inflorescence architecture: the indeterminate and the determinate. The main axis of indeterminate plants develops infinitely and only produces blooms on its sides. The main axis of plants with a determinate inflorescence is finite, and the shoot apical meristem transforms into a flower. Almost all Cucurbitaceae crops are indeterminate plants. The flower occurrence is influenced by the growth of the main axis. A non-synonymous SNP in *Terminal Flower1 (CsTFL1)* causes a determinate growth habit, according to map-based cloning of the determinate (*de*) locus [92]. *CsTFL1* knockdown results in the determinate growth and the development of terminal flowers in cucumber. *CsTFL1* has been found to compete with *Flowering Locus T*
*(**CsFT**)* for interaction with *Negative on TATA less2-FD Paralog (CsNOT2a-CsFDP)* to impede the determinate development and terminal flower formation. *LEAFY* (*CsLFY)* interacts with *CsWUS* to regulate shoot meristem maintenance and flower development in cucumber via activating *APETALA3* (*CsAP3)* and *Cucumber MADS box gene 1* (*C**s**CUM1). CsLFY-RNAi* transgenic cucumbers exhibit a transformation from indeterminate to determinant growth [93]. In addition to the determinate growth phenotype, some dwarf mutants have also been identified in cucumber, such as *Cucumber dwarf*
*(**Csdw**),*
*compact*
*(**cp**),*
*compact-1*
*(**cp-1**),*
*super compact-1*
*(**scp-1**),*
*super compact-2*
*(**scp-2**),* and *short internode*
*(**si**)* [39,40,94,95]. The *Csdw* mutant exhibits short internodes due to decreased endogenous GA3 levels and decreased cell counts in the main stem (Figure 3).

## 7. Conclusions and Perspectives

Recent advances in research have extended our understanding of how phytohormones and genes influence plant organ sizes and shapes. The main purpose of plant breeding is to develop plants with excellent agronomic and economic qualities and yields, which involves enhancing the leaf and fruit outputs of many plants. A subset of plants may alter the geometry of their leaves and the angle established between the leaf and the main stem to optimize light energy uptake, resulting in better photosynthetic yields and biomass outputs. However, many difficult questions need to be solved in order to understand how plant size is controlled. Cells and organs both have different sizes. How we accomplish the complex calculation and execution of the correct sizes is both an intellectually compelling and critical crop science problem. For example, leaves are the main site for photosynthesis, which provide energy for crop growth and are closely related to the yield. Larger leaves are more beneficial in improving photosynthetic efficiency. However, in the limited cultivation space, reasonably close planting requires smaller leaves, which is also very important in production. Therefore, how we accurately control the leaf size to achieve optimal production is still a problem to be solved. Shoot branching is also an important agronomic trait that directly determines the plant architecture and affects crop productivity. In production, axillary branches need to be manually removed to promote crop yields and qualities for the fresh market, especially in cucumber. Mutants with less branches can reduce labor costs significantly. However, less branches greatly affect the yields of the towel gourd and pumpkin, which are mainly produced from lateral branches. Therefore, how we balance the relationship between the number of lateral branches and the yield remains a question to be studied.

Despite the guidance that can be derived from the genetic information contained in each plant cell, phytohormone levels and phytohormone signaling pathways are also required for organ development and morphogenesis. Moreover, the roles of different phytohormones are not separate entities, but instead partially overlap. For example, both auxins and CKs are indispensable for SAM development and maintenance, but they play distinct roles. The formation and organogenesis of the leaf require auxins, while CKs promote meristem maintenance. However, they do not exist or function independently from the phytohormones. Many studies have revealed that auxins and CKs interact in a variety of cells, tissues, and organs in both antagonistic and synergistic ways [96,97,98]. Therefore, a more thorough and precise understanding of plant development can be obtained by studying the interactions among phytohormones.

A great deal of research over the last few decades has considerably enhanced our understanding of plant organ development mechanisms by employing a range of biotechnological approaches, most frequently in the model plant *Arabidopsis*. At present, many genes regulating the size and shape of cucumber fruits have been identified, but the specific mechanism of action has not been thoroughly studied. The rapid expansion period of cucumber fruit cells is an important factor, leading to the rapid growth of the fruits. At present, only through transcriptome or fluorescence quantitative analysis has it been discovered that some cell expansion genes are upregulated during the rapid expansion period of fruits or participate in the development of cucumber fruits and affect the shape of the fruits [99]. Further study of the genes related to fruit cell expansion and their regulatory mechanisms is required [99]. Many genes regulating leaf development, such as *TCP (TEOSINTE BRANCHED1/CYCLOIDEA/PCF)*, *KNOX (KNOTTED1-like homeobox)*, and *ANT(ANTEGUMENTA)* exist and have been identified in *Arabidopsis*, tomato, and other crops [100,101,102]. *miRNA* is also reported to regulate the leaf size [103]. However, there are few reports or in-depth studies on cucumber, or even on cucurbitaceae crops. Furthermore, additional receptors of certain essential genes in cucurbits have yet to be found. To fully comprehend the physiological mechanisms underlying plant organ development, various components and their interactions must be examined holistically rather than separately.

With the advancement of molecular breeding methods, crop agriculture features may now be modulated by selecting or modifying target genes. Identifying and understanding the organ size regulators in cucumber can help to enhance Cucurbitaceae crops through the breeding of high-yield crops with perfect organ sizes, as well as unique vegetable types with ideal leaf and plant sizes.

## Figures and Tables

**Figure 1 plants-11-02128-f001:**
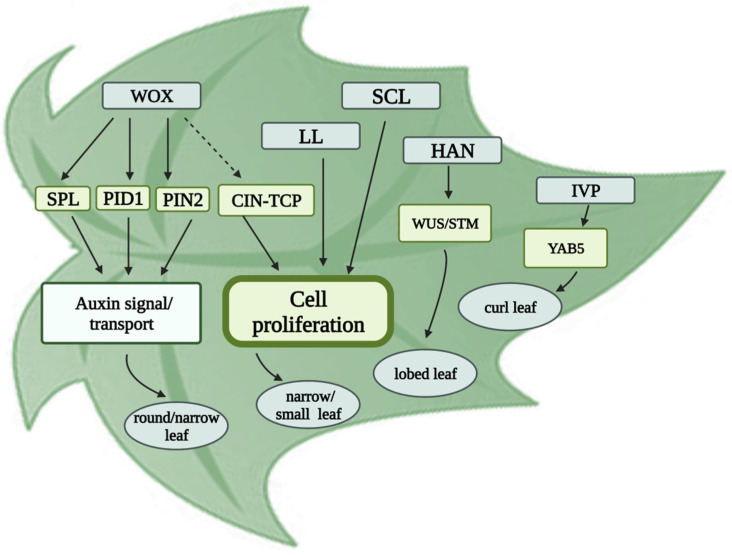
Genetic and molecular regulation mechanisms of cucumber leaf morphology. The grey box indicates the genes/proteins related to leaf morphology identified in cucumber. The yellow box indicates the gene or protein regulated by the target gene. The gray oval box indicates the corresponding leaf phenotype. The arrows indicate the regulatory relationship between the two genes. Abbreviations in figure, WOX: Wuschel-related homeobox; LL: little leaf; SCL: small and cordate leaf; HAN: Hanaba Taranu; IVP: irregular vasculature patterning; SPL: SPOROCYTELESS; PID1: PINOID 1; PIN2: PIN-FROMED 2; STM: Shoot Meristemless).

**Figure 2 plants-11-02128-f002:**
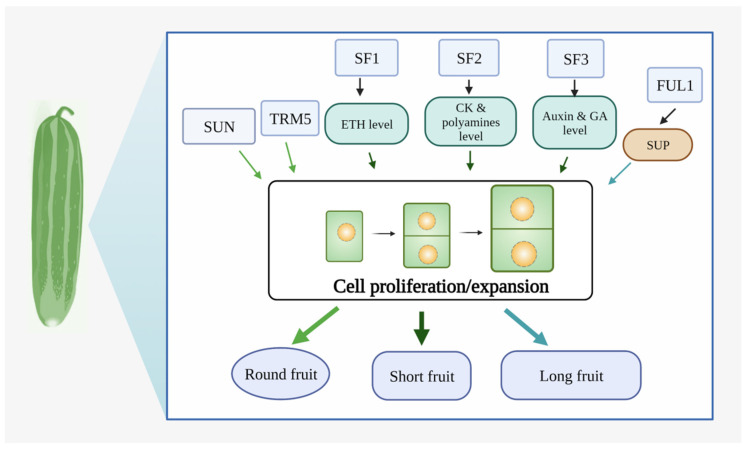
Genetic and molecular regulation mechanisms of cucumber fruit development. The grey box indicates the genes/proteins related to fruit development identified in cucumber. The blue-green boxes indicate hormonal pathways. SUN and TRM5 regulate round fruit by cell proliferation. SF1, SF2, and SF3 regulate short fruit by cell proliferation via the levels of ETH, CK/polyamines, and auxin/GA, respectively. CsFUL1 targets SUP and negatively regulates fruit elongation by cell proliferation and expansion. Abbreviations in figure, TRM5: Tonneau1 recruiting motif 5; SF1/2/3: Short Fruit; FUL1: Fruitfull 1; SUP: Superman.

**Figure 3 plants-11-02128-f003:**
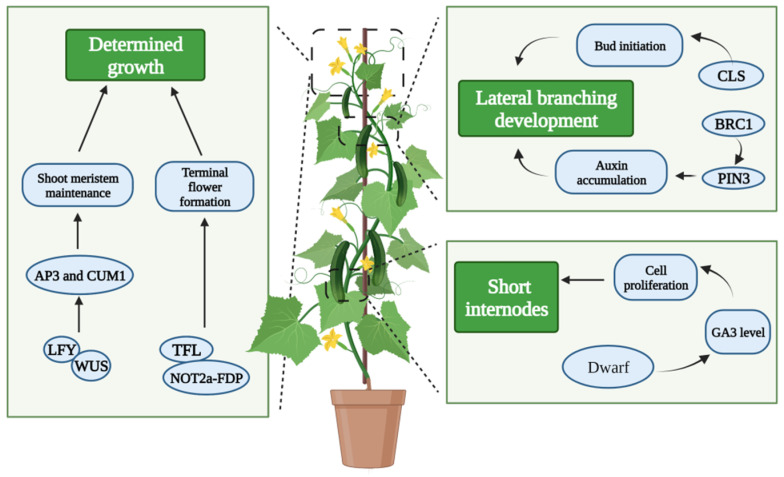
Genetic and molecular regulation mechanisms of the cucumber plant architecture. TFL1’s interaction with CsNOT2a-CsFDP impedes terminal flower formation, and LFY interacts with CsWUS to regulate shoot meristem maintenance in cucumber via activating CsAP3 and CsCUM1 to regulate cucumber towards a determined growth pattern. BRC1 regulates lateral branching by influencing auxin accumulation, while CLS influences bud initiation. *Dwarf* regulates the internodes by the regulation of endogenous GA3 levels and cell counts in the main stem. Abbreviations in figure, LFY: LEAFY; AP3: APETALA3; CUM1: Cucumber MADS box gene 1; TFL: Terminal Flower1; NOT2a-FDP: Negative on TATA less2-FD Paralog; BRC1: Branched1; CLS: Cucumber Lateral Suppressor.

## Data Availability

All the data are included in the present study.

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
