# Peer review of "Research Progress on the Leaf Morphology, Fruit Development and Plant Architecture of the Cucumber"

_plants, 2022, doi:10.3390/plants11162128_

Round 1

Reviewer 1 Report

The knowledge of the Genetic and Molecular Regulation of Leaf and Fruit development has been mainly studied on Arabidopsis and some research has been performed on agronomic species. The present work is not only concentrated on cucumber it is more a general contribution to the understanding of the genetics of leaf morphology and fruit development until 2020.  The work needs to include more recent references (200-2022). One problem is the title because the content is not only focused on cucumber only the latest sections (5 and 6) are dedicated to cucumber. I suggested eliminating the word “cucumber” from the title.

Section 6.3 The Roles of Phytohormones on Growth and Development

Is a comprehensive review of the role of phytohormones on the size of a plant organ development. The authors include examples and explanation of the five classical phytohormones [auxin, CK, ethylene (ETH), gibberellin (GA), and abscisic (ABA)], the small molecules such as brassinosteroid (BR), jasmonic acid (JA), salicylic acid (SA), and strigolactone (SL) in plant growth.  The content is focused on CK, ETH, and GA but ABE, BR, SA, and SL need more attention. (p. 3-5).

Then the authors go into the 4. Genetic and Molecular Regulation Mechanisms of Cucumber Leaf Morphology (p.5). The authors organized the information into three sections

1.     PINOID (PID) belongs to the plant-specific AGCV III subfamily of the AGC protein 193 kinases large family, which mediates phosphorylation/dephosphorylation by encoding 194 serine/threonine protein kinases.

2.     WOX 215 , Wuschel-related Homeobox (WOX) genes, belong to the Homeodomain (HD) a superfamily 216 which has a typical DNA-binding domain of ~60 amino acids.

3.     (3) Other genes. In Arabidopsis, Hanaba Taranu (HAN) encodes a GATA3-type transcription factor that is involved in floral organ development, shoot apical meristem (SAM) organization, and embryo development.

The content of this section is not exclusive to Cucumber, the references used to support the investigation mainly use Arabidopsis and other species.  My recommendation is to eliminate the word “Cucumber” in the subtitle. 4. Genetic and Molecular Regulation Mechanisms Leaf Morphology

p. 7 lines 296-299 . “Generally, cucumbers have simple fruit shapes (round, oblong/oval, or cylindrical). Different genotypes of fruits can grow to specific size and form, implying that genetic variables are the most important determinants of final size and shape (Gillaspy et al. 1993; 298 Tanksley 2004; Colle et al. 2017)”. The sentence is to general, to be included as information for Cucumber. Tanksley 2004 and Gillaspy et al. 1993 used Tomato as a model; Colle et al. 2017 publish on cucumber, Gillaspy et al. 1993

The work is well organized and comprehensively described.

The manuscript is organized in the following sections: 1. General introduction on 2. Cell Proliferation and Expansion in Plant Growth and Development, 3. The Roles of Phytohormones on Growth and Development, 4. Genetic and Molecular Regulation Mechanisms of Cucumber Leaf Morphology 5. Genetic and Molecular Regulation Mechanisms of Cucumber Fruit Development, 6. Genetic and Molecular Regulation Mechanisms of Cucumber Plant Architecture, 7. Conclusion and Perspectives.

Section 4.2.2 Molecular Regulation Mechanisms of Fruit Size is in section 5.1. Molecular Regulation Mechanisms of Fruit Shape. It needs correction

I agree with the arrangements of sections, but sections 5, 6, 7, do not refer only to Cucumber, and they included information on genes and mutants of tomato, Arabidopsis, cucumber, and other species. When referring to Cucumber in a sentence, be careful to place only references to cucumber.

Considering that each figure should be explained by itself the foot, each figure should include the explanation of the abbreviations in full.

The work is supported by scientific references, from recognized journals.

But the title “Research Progress on the Leaf Morphology, Fruit Development and Plant Architecture of cucumber “

It is not clear if the paper refers to Leaf morphology and Fruit Development in general and besides more extended information on cucumber. If this is the case the title should be clearer.

Are there appropriate and adequate references to related and previous work?

The references are appropriate and adequate, but they need to be updated.

Ther are some adequate and recent references:

But most of the references are from 2003 to 2018 and are not specific to Cucumber.

For the sections 1,2,3, and 4, which are more general, please add more recent references

The references in sections 5 and 6 are more specific to cucumber, and they represent a group of research that follows a line of research, according to the subject of the submitted manuscript.

The document is readable and there are a few mistakes in English grammar and syntax, but they are minor mistakes and are indicated in the PDF document.

Reviewer 2 Report

The manuscript by Li et al. tried to probe the progresses on the Leaf morphology, fruit development and plant architecture of cucumber, and it was written well. But for some parts of this review, there is not too much progress, and most of the part is describing the results from other plants. Therefore, the authors should focus on the specific aspects that really obtained enormous progress.

Reviewer 3 Report

Dear authors,

thank you for your submission. I enjoyed the read!  You laid out the purpose for the review clearly, and work through the different sections in detail. 

I think that some minor revisions for grammatical issues should be addressed, but the science appears sound and the writing style and content are well done. I think that the review you provide will help direct future work to develop cucumbers with optimally sized organs for enhancing yield.

I have suggested some minor edits, which you can find in the copy of the manuscript I've attached. I've noted that I couldn't find any in-text figure references, so please add these if they aren't there (I've explained more in the attachment). I have also indicated where the work can use the support of additional citations, so please add these where I've indicated.

Thank you again and have a great day!
